# Mitigation of Acute Hydrogen Sulfide and Ammonia Emissions from Swine Manure during Three-Hour Agitation Using Pelletized Biochar

**Baitong Chen** [1], **Jacek A. Koziel** [1,*], **Myeongseong Lee** [1], **Samuel C. O'Brien** [1], **Peiyang Li** [1] **and Robert C. Brown** [2]

[1] Department of Agricultural and Biosystems Engineering, Iowa State University, Ames, IA 50011, USA; baitongc@iastate.edu (B.C.); leefame@iastate.edu (M.L.); scobrien@iastate.edu (S.C.O.); peiyangl@iastate.edu (P.L.)

[2] Department of Mechanical Engineering, Bioeconomy Institute, Iowa State University, Ames, IA 50011, USA; rcbrown3@iastate.edu

[*] Correspondence: koziel@iastate.edu

**Abstract:** The risk of inhalation exposure to elevated concentrations of hydrogen sulfide ($H_2S$) and ammonia ($NH_3$) during the agitation of stored swine manure is high. Once or twice a year, farmers agitate manure before pump-out and application to fields. Agitation of the swine manure causes the short-term releases of highly toxic levels of $H_2S$ and $NH_3$. In our previous pilot-scale studies, the biochar powder showed significant mitigation of $H_2S$ and $NH_3$ emissions when it was surficially applied to manure immediately before agitation. However, fine biochar powder application poses hazards by itself and may not be practical to apply on a farm scale, especially when livestock and workers are present. We hypothesized that applying pelletized biochar to manure surfaces is just as effective as applying powder to protect farmers and animals from excessive exposure to $H_2S$ and $NH_3$. This work reports on the lab-scale proof-of-the-concept trials with biochar pellets on the lab scale. The objective was to compare the biochar pellets and biochar powder on their effectiveness of mitigation on $H_2S$ and $NH_3$ gases during 3-h-long swine manure agitation. Three scenarios were compared in ($n = 3$) trials: (i) control, (ii) 12.5 mm thick surficial application to manure surface of biochar powder, and (iii) an equivalent (by mass) dose of pelletized biochar applied to the manure surface. The biochar powder was bound with 35% (wt) water into ~5 × 10 mm (dia × length) pellets. The biochar powder was significantly ($p < 0.05$) more effective than the biochar pellets. Still, pellets reduced total $H_2S$ and $NH_3$ emissions by ~72% and ~68%, respectively ($p = 0.001$), compared with ~99% by powder ($p = 0.001$). The maximum $H_2S$ and $NH_3$ concentrations were reduced from 48.1 ± 4.8 ppm and 1810 ± 850 ppm to 20.8 ± 2.95 ppm and 775 ± 182 ppm by pellets, and to 22.1 ± 16.9 ppm and 40.3 ± 57 ppm by powder, respectively. These reductions are equivalent to reducing the maximum concentrations of $H_2S$ and $NH_3$ during the 3-h manure agitation by 57% and 57% (pellets) and 54% and 98% (powder), respectively. Treated manure properties hinted at improved nitrogen retention, yet they were not significant due to high variability. We recommend scaling up and trials on the farm-scale level using biochar pellets to assess the feasibility of application to large manure surfaces and techno-economic evaluation.

**Keywords:** agricultural wastewater treatment; hazardous waste management; gas–liquid interface; gaseous emissions; animal production; occupational safety

## 1. Introduction

Animal production provides an excellent source of fertilizer in the form of manure for crop production. The animal production system's sustainability depends on the efficient carbon (C) and nitrogen (N) cycling between manure, crops, and animal feed. Swine manure is generated and stored year round. However, manure can be utilized by crops seasonally, typically before seed planting and/or after the harvest.

Stored manure requires pump-out and emptying to make room for a continued farm operation. Short-term (hours) agitation of manure is required to stir up and incorporate settled solids into a slurry that pumps can handle. Manure agitation is facilitated by using a high-capacity pump with recirculation for a vigorous mechanical stirring of settled manure solids at the bottom of the storage pit.

Agitation breaks the entrapped gas bubbles and causes the acute release of toxic gases (hydrogen sulfide; $H_2S$) and nutrients (ammonia; $NH_3$) [1–3] from swine manure. This routine procedure sometimes results in the loss of livestock and rare unfortunate incidents involving human life loss due to excessive inhalation of $H_2S$. The Occupational Safety and Health Administration (OSHA) recommends the permissible exposure limits (PELs) concentration for $H_2S$ at 20 ppm (General Industrial Peak Limit) and an acceptable maximum peak above the acceptable ceiling concentration at 50 ppm (General Industry Ceiling Limit), with a maximum duration of 10 min [4]. Hoff et al. (2006) reported the maximum measured $H_2S$ concentrations of 36 ppm and 16 ppm in the pit exhaust and tunnel (barn) fans, respectively, during swine manure agitation of a deep-manure storage farm in Iowa [3]. The average $H_2S$ concentrations during agitation were 18 and nearly 28 times higher in the pit exhaust and tunnel and sidewall exhaust fans, respectively, compared with the before-removal levels [3]. Chénard et al. (2003) reported $H_2S$ concentrations as high as 1000 ppm during plug-pulling events in shallow manure storages [5]. The $H_2S$ concentrations exceeded 100 ppm in as many as ~30% of the plug-pulling events [5]. In addition, Chénard et al. (2003) reported that a power washing could agitate manure and increase the $H_2S$ concentrations in the workers' environment over the 15 ppm in approximately 18% of events monitored. Pesce et al. (2007) reported that the $H_2S$ concentrations in confined space over manure storage could be mitigated below the OSHA PELs with site-specific ventilation strategies [6]. Ni et al. (2018) studied the formation and release of gas (including $H_2S$) microbubbles in manure storage and during and post agitation [7]. The 'bubble-release' model explained the '$H_2S$ burst-release' phenomenon reported in mechanically ventilated swine buildings [7].

The $NH_3$ gas is one of the causes of odor and secondary particulate matter (PM2.5) aerosols affecting the surrounding communities' air quality. The U.S. National Institute for Occupational Safety and Health (NIOSH) recommends the time-weighted average (TWA) 10-h concentration for $NH_3$ at 25 ppm and a short-term exposure limit (STL) of 15 min at 35 ppm [8]. Alvarado et al. (2019) reported that $NH_3$ concentrations exceeded the TWA limit for 11 out of 50 monitoring days due to tasks such as weighting, feeding, and draining manure pits [9]. Hoff et al. (2006) reported the 4–5 times increase in maximum measured $NH_3$ emissions during swine manure agitation compared to before agitation [3]. CIGR (1992) and Busse (1993) reported that 20 to 40 ppm of $NH_3$ would increase respiratory diseases, 50 to 150 ppm causes a decrease of pig growth by 12–29%, and 100 to 200 ppm could cause irritation and anorexia to swine and workers [10,11].

Nour et al. (2021) reported 133 manure-related incidents from 1976 to 2019 in the seven mid-western U.S. states [12]. Most of the fatalities (57%) were caused by suffocation or asphyxiation from the toxic gases emitted from manure [12]. Mitloehner and Calvo (2008) raised awareness that the number of incidents related to toxic emissions could be underestimated and not well documented [13]. In addition, the health impact of the mixture of toxic gases emitted from manure is also poorly understood [13].

There is no proven, widely adopted technology to mitigate the risk and gaseous emissions from agitated manure. Farmers take precautions by maximizing ventilation during agitation and generally avoiding being near the agitated manure storage. However, animals can still be exposed to acute releases of uncontrolled emissions. The unfortunate loss of human and animal life continues to occur yearly. Predicala et al. (2007) reported that implementing a manure scraper system in a swine grower-finisher shallow barn could reduce $H_2S$ concentrations by ~90% while the $NH_3$ concentration increased by 36% [14]. Alvarado and Predicala (2017) reported that using a fluidized bed air filtration system loaded with zinc oxide nanoparticles can reduce ~65% of $H_2S$ and 42% of $NH_3$ in a farm-

scale trial [15]. However, the cost of implementing these types of technology is relatively high (>CAD $6/pig). Chen et al. (2018) showed that iron oxide could reduce up to 94% of $H_2S$ production from gypsum-laden dairy manure on a lab scale [16]. In addition, Chen et al. (2018) reported that the $H_2S$ releases were largely enhanced by agitation and proposed iron oxide treatment during agitation events [16]. Researchers have shown the effective $H_2S$ and $NH_3$ reduction with calcium hydroxide, sodium nitrate, and hydrogen peroxide treatments at the lab scales, but the potential of scaling up to real farm application was limited by the economics [17–19]. A recent comprehensive evaluation of marketed manure additive products for controlling odor emissions and nutrient losses did not show statistically significant mitigation of $H_2S$ and $NH_3$ [20]. Thus, there is a need to mitigate the safety concerns and gaseous emissions representing the loss of nutrients and manure value as a fertilizer.

We have been advancing the use of carbon-rich adsorbent (biochar) as a manure treatment to mitigate gaseous emissions from stored manure [21]. Biochar is a by-product of the thermal processing of biomass. Biobased-fuel production, waste-to-carbon, and waste-to-energy thermal processes result in a relatively low-value biocoal. Circular economy opportunities exist for the valorization of biochar and the improvement of sustainability in animal and crop production systems [22,23]. The long-term goal is to test and scale up the treatments from the laboratory scale to the farm scale, keeping in mind the techno-economic constraints for many swine farmers.

Biochar powder has been proposed as a soil amendment, and an adsorbent with properties similar to activated charcoal, fertilizer, and alternative fuel [24]. Biochar can be made via pyrolysis or torrefaction from biomass and waste, which, with different feedstock and process conditions, resulted in biochar with different physicochemical properties [24].

Biochar powder effectively mitigates emissions of $H_2S$ and $NH_3$ for both long-term (weeks to months) [21,25] and short-term (few minutes) trials [26,27]. Chen et al. reported an up to 60% reduction in $H_2S$ emissions and 70% to 80% reduction in $NH_3$ emissions by surficial biochar powder application immediately before 3-min manure agitation [26,27]. Thus, synergistic effects to biochar use could be achieved for the animal-crop production system. First, biochar can be used to mitigate gaseous emissions from manure, and then the biochar and manure mixture can be used as a better-quality fertilizer, improve the soil nutrients content, and minimize the nutrient losses from soil [22,23]. Therefore, innovative biochar treatment could be a one-stop solution to solve the gaseous emissions and improve agriculture's sustainability.

It is impractical to apply biochar powder to large manure surfaces on a farm scale. Handling powder might be hazardous, especially when livestock and workers are present. Biochar is a fine and lightweight powder that can generate PM air pollution and potentially self-ignite [28], while another fuel (methane, $CH_4$) is continuously generated by swine manure.

Thus, we propose to use pelletized biochar for the treatment of gaseous emissions from agitated manure. The corn- and wood-derived biochars are difficult to pelletize; therefore, a low-cost 'additive' ('glue') to hold them together is needed. Socio-economic constraints also drive the selection of additives to make biochar pellets for this purpose. However, no data exist on how the pelletized biochar behaves when applied to manure and how effective it is in mitigating gaseous emissions, particularly $H_2S$ and $NH_3$.

This research aimed to compare the biochar pellets and biochar powder on their effectiveness in mitigating $H_2S$ and $NH_3$ gases during swine manure agitation. A 3 h agitation was used to represent a realistic timeline for manure storage, stirring immediately before and during the field application.

## 2. Materials and Methods

The experiment evaluated three scenarios with $n = 3$ replication during 3 h of manure agitation: (i) control, (ii) 12.5 mm ($\frac{1}{2}$ inch) thick surficial application to manure surface of biochar powder, and (iii) and an equivalent (in mass) dose of pelletized biochar applied

to the manure surface. The previous research showed that both 12.5 mm and 6 mm thick surficial application has a similar effect in reducing emissions [26,27]. This research selected the thicker powder dose (12.5 mm) to minimize surficial coverage bias due to the pelletized biochar's corresponding dose. The pelletized biochar is densified and cannot cover the manure surface as efficiently as uniformly spread powder. This decision was also made to accommodate the proof-of-concept trial that biochar pellets can still effectively mitigate emissions.

The reactor used in this experiment has a height of ~40 cm (15 in) and a diameter of 14 cm (5.5 in) (Figure 1). The working volume of manure was 3 L, freshly collected from a local swine farm with a deep pit and stored in the lab during the experiment. The Masterflex L/S pump (Masterflex, Gelsenkirchen, Germany) recirculated the manure at a rate of 10 mL/s. The system was designed to simulate swine manure agitation. However, this lab-scale study's hydraulic retention time was much smaller than the swine manure agitation on a farm [29]. At the end of this study, 18 samples (3 scenarios with $n = 3$ and before and after agitation) of manure samples were sent to Brookside Laboratories, Inc. (New Bremen, OH, USA) for analyses. The airflow rate was kept at 2 L/min to match the requirements of the gas measurement system. The feedstock of the HAP biochar was corn stover and was pyrolyzed at 500 °C. Corn stover was ground to 3 mm and treated with 8% (wt) iron sulfate resulting in a pH of 5.2.

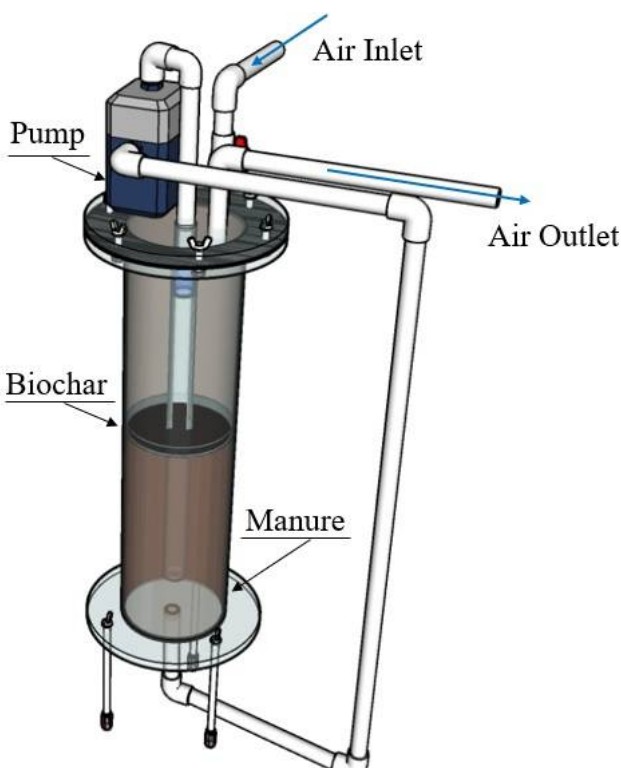

**Figure 1.** The schematic of the reactor to simulate the process of manure agitation before pump-out.

## 2.1. Pelletization of Biochar

In this research, mild acidic (pH 5.2) Fe-treated porous autothermal (HAPE) biochar made from corn stover was bound with 35% (wt) water into ~5 × 10 mm (3/16 × 3/8 inch, dia × length) pellets. The biochar powder was mixed with 35% (wt) water, then feed into pellet mill PMCL5 (California Pellet Mill, Crawfordsville, IN, USA) with ~5 mm disks. The length of pellets varied, but the average was around ~10 mm. The effect of the pelletization process on the biochar morphology was then visually compared using scanning electron microscopy (SEM).

### 2.2. NH₃ and H₂S Measurements

An OMS-300 (Smart Control and Sensing Inc., Daejeon, Korea) real-time monitoring system equipped with electrochemical gas sensors $H_2S$/C-50 and $NH_3$/CR-5000 (Membrapor, Wallisellen, Switzerland) was used to measure the real-time $H_2S$ and $NH_3$ concentration in units of parts per millions (ppm). The $H_2S$ sensor can measure up to 100 ppm, and the $NH_3$ sensor can measure up to 5000 ppm. All sensors were calibrated with standard gases before the experiment [26,27]. The real-time monitoring system sampled the outlet gas of the reactor continuously at 2 L·min⁻¹. Measured concentrations were recorded every minute. Care was taken to minimize the risk of sensor overload and signal drift.

### 2.3. Manure Properties

In total, 18 samples (3 scenarios with $n = 3$ and before and after agitation) of manure samples were sent to Brookside Laboratories, Inc. (New Bremen, OH, USA) for analyses. The manure property data were comprehensive (i.e., including moisture, total nitrogen, ammonium-N, nitrate-N, organic N, and metals content) following the 'Recommended Methods of Manure Analysis (RMMA)' and 'Test Methods for the Examination of Composting and Compost (TMECC)' [30,31]. Both procedures provide benchmark methods for compost analysis to enable the comparison of analytical results. Each parameter is presented in its section and generally includes more than one protocol or test method. In addition, the manual contains more parameters that might be of concern or interest for a particular situation. For example, moisture/dry matter followed Method 2.0 from RMMA; total nitrogen followed Method 3.3 from RMMA and TMECC method 4.02. Total carbon followed Method 4.01 from TMECC. Ammonium-N followed RMMA method 4.3. Nitrate-N followed the guideline of Environmental Protection Agency (EPA) 353.2. Mineral digestion was used to determine phosphorous (P), potassium (K), calcium (Ca), magnesium (Mg), sodium (Na), and sulfur (S) followed Method 5.3 from RMMA and TMECC 4.12 [30–32].

Changes in manure properties (Δ Control, Δ Pellets, and Δ Powder) were calculated using the following Equation (1).

$$\Delta \text{ manure property} = \text{manure property after} - \text{manure property before} \tag{1}$$

Manure properties such as moisture, mineral matter, and various chemicals are all in the units of % wet basis. Then manure properties were compared using Δ Control, Δ Pellets, and Δ Powder.

The percent differences (*%Diff*) were calculated based on Δ Control vs. Δ Pellets and Δ Control vs. Δ Powder. The following Equation (2) was used:

$$\%Diff = \frac{\Delta \text{ Treatment} - \Delta \text{ Control}}{\text{Abs}(\Delta \text{ Control})} * 100\% \tag{2}$$

where the Δ Treatment is either the changes in manure properties of biochar powder or pellets (Δ Pellets or Δ Powder) and Abs (Δ Control) is the absolute value of the Δ Control. Thus, a negative *%Diff* indicates manure properties of Δ Treatment where *%Diff* is less than the manure properties of Δ Control; a positive *%Diff* indicates otherwise.

### 2.4. Data Analysis

The $H_2S$ and $NH_3$ concentrations (ppm) were converted into flux units of mg/min using the standard lab conditions. Then, the percent reductions (%R) of biochar treatments were calculated with Equation (3):

$$\%R = \frac{E_{Control} - E_{biochar}}{E_{Control}} * 100\% \tag{3}$$

where $E_{control}$ is the average total emissions or max concentration of $H_2S$ or $NH_3$ from manure without any treatment during manure agitation, and $E_{biochar}$ is the average total

emissions or max concentration of $H_2S$ or $NH_3$ from manure treated with biochar powder or pellets during agitation.

The one-way analysis of variances (ANOVA) and Tukey–Kraner method were used to analyze the emissions and calculated the *p*-values of associated %R. The one-tailed T-test was used to calculate the *p*-values for the %R of the max concentrations. All data analyses were done in JMP (version Pro 15, SAS Institute, Inc., Cary, NC, USA). A *p*-value less than 0.05 was used to determine significance.

## 3. Results

A nearly immediate spike in the headspace concentrations of $NH_3$ and $H_2S$ was observed at the start of agitation for all treatments. This spike was also the highest concentration throughout each trial. After the initial spike, the gas concentrations stabilized with a slightly decreasing trend for the pellet treatment (Figures 2 and 3) or returned to the pre-agitation concentrations (for the powder treatment).

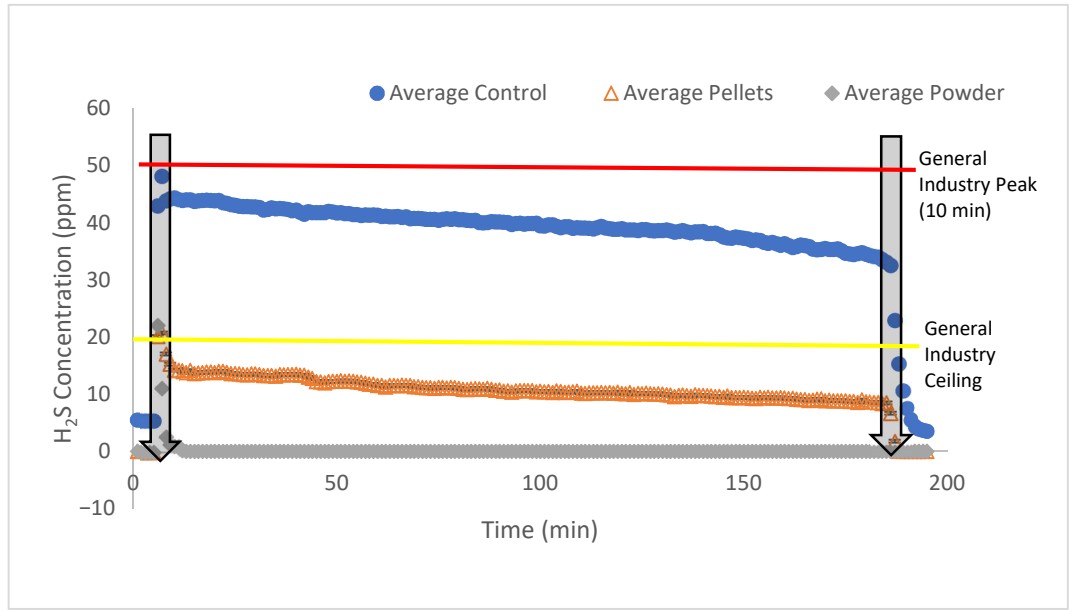

**Figure 2.** The average headspace concentrations of $H_2S$ during the 3 h agitation of swine manure treated with biochar pellets and powder. The vertical arrows indicate the start and end of manure agitation. The red and yellow lines indicate the OSHA General Industry Peak (50 ppm) and the General Industry Ceiling (20 ppm) concentrations of $H_2S$.

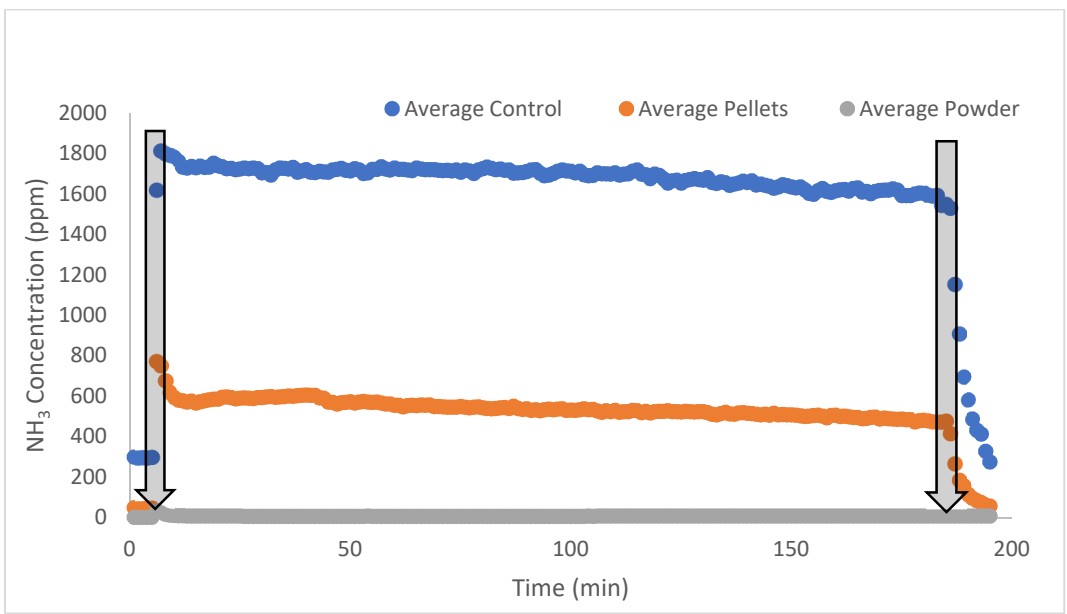

**Figure 3.** The average headspace concentrations of $NH_3$ during the 3 h agitation of swine manure treated with biochar pellets and powder. The vertical arrows indicate the start and end of manure agitation. The U.S. National Institute for Occupational Safety and Health (NIOSH) recommends the time-weighted average (TWA) 10-h concentration for $NH_3$ at 25 ppm and a short-term exposure limit (STL) of 15 min at 35 ppm (not shown to retain figure legibility).

The biochar powder had a greater mitigation effect than biochar pellets for both $H_2S$ and $NH_3$ emissions during manure agitation (Table 1). The biochar pellet treatment significantly reduced the maximum $H_2S$ concentration by 57% ($p = 0.02$). The $NH_3$ maximum concentration was also reduced by the same percentage (57%), but it was not significant ($p = 0.08$). On the other hand, the biochar powder treatment significantly reduced the maximum $NH_3$ concentration by 98% ($p = 0.001$), while the percent reduction of the maximum $H_2S$ concentration (54%) was not significant ($p = 0.052$).

**Table 1.** The maximum headspace concentrations of $H_2S$ and $NH_3$ (±standard deviation) and % reductions (%R) during the 3 h agitation of swine manure averaged for $n = 3$ trials. Statistical significance of %R is reported as ($p$-values) and bold font. The letter of the groups resulting from the ANOVA indicates significant differences.

|  | Control | Pellets | Powder |
|---|---|---|---|
| Maximum $H_2S$ Concentration, ppm | 48.1 ± 4.84 | 20.8 ± 2.95 | 22.1 ± 16.9 |
| (letter of groups) | (A) | (B) | (B) |
| %R | - | **57%** | 54% |
| ($p$-value) |  | (0.02) | (0.052) |
| Maximum $NH_3$ Concentration, ppm | 1811 ± 852 | 775 ± 182 | 40.3 ± 57.0 |
| (letter of groups) | (A) | (B) | (C) |
| %R | - | 57% | **98%** |
| ($p$-value) |  | (0.08) | (0.001) |

The total $H_2S$ and $NH_3$ emissions were also significantly ($p = 0.001$) mitigated by biochar pellets and powder (Table 2). Biochar pellets reduced the total $H_2S$ and $NH_3$ emissions by 72% and 68%, respectively, whereas biochar powder reduced the $H_2S$ and $NH_3$ by 99% during the 3 h of swine manure agitation.

**Table 2.** The total emissions of $H_2S$ and $NH_3$ (±standard deviation) and % reductions (%R) during the 3 h agitation of swine manure averaged for $n = 3$ trials. Statistical significance is reported as (*p*-values) and bold font. The letter of the groups resulting from the ANOVA indicates significant differences.

|  | Control | Pellets | Powder |
|---|---|---|---|
| Total Emission of $H_2S$, mg/m$^2$ | $1.31 \pm 0.305$ | $0.361 \pm 0.0453$ | $0.0071 \pm 0.005$ |
| (letter of groups) | (A) | (B) | (C) |
| %R | | **72%** | **99%** |
| (*p*-value) | - | (0.001) | (0.001) |
| Total Emission of $NH_3$, mg/m$^2$ | $28.0 \pm 12.3$ | $8.93 \pm 1.70$ | $0.152 \pm 0.216$ |
| (letter of groups) | (A) | (B) | (C) |
| %R | | **68%** | **99%** |
| (*p*-value) | - | (0.001) | (0.001) |

The average $H_2S$ concentrations of the control and both treatments were lower than the General Industry Peak (50 ppm, 10 min), but the $H_2S$ concentrations of the control were above the General Industry Ceiling (20 ppm) during the 3-h agitation. Both biochar pellet and powder treatments kept the $H_2S$ concentrations below the General Industry Ceiling except for the first two minutes of agitation (Figure 2).

The average $NH_3$ concentration for the biochar powder treatment was below both TWA and STL limits (25 and 35 ppm, respectively). In contrast, the average $NH_3$ concentrations of the control and biochar pellet treatment were above STL and TWA limits even before the manure agitation and throughout the experiments.

All manure and biochar pellet- and powder-treated manure are summarized in the Supplementary Materials. There was no statistical significance to the selected manure properties due to the high variability in manure properties changes among the control, pellets, and powder (Table 3, Table A1) except for the mineral matter and Na content. It is worth noting that both biochar pellets and powder (−0.008) showed ~50% smaller decreases in total N (−0.008% and −0.007%, respectively) when compared with the control (−0.016), indicating a potential to retain more N in manure with biochar treatments. In addition, biochar pellets and powder showed a smaller decrease in ammonium-N (~64%, −0.005 and ~45%, −0.008, respectively) than the control (−0.015). Both biochar treatments increased the carbon content in the manure, whereas the control showed a decrease in carbon content, as shown by the carbon % and C/N ratio (Table 3, Table A1). Additional work and scale-up trials are still needed to elucidate statistical significance to these initial observations hinting at the potentially improved manure quality treated with biochar.

**Table 3.** The average changes (Δ) in manure properties from 'before' and 'after' 3-h manure agitation. A negative Δ value indicates a decrease, and a positive value indicates an increase. Moisture, mineral matter, and targeted chemicals are in units of % wet basis. A negative *%Diff* indicates manure properties of Δ Treatment were *%Diff* less than the manure properties of Δ Control; a positive *%Diff* indicates otherwise. The bold font indicates statistical significance. The *p*-values are listed inside parentheses.

| Manure Property | Δ Control (% Wet Basis) | Δ Pellet (% Wet Basis) | Δ Powder (% Wet Basis) |
|---|---|---|---|
| Moisture | 0.073 | 0.043 | −0.020 |
| *%Diff* | - | −41 (0.41) | −127 (0.07) |
| Mineral Matter | −0.067 | −0.053 | 0.010 |
| *%Diff* | - | 20 (0.43) | **115 (0.03)** |
| Total Nitrogen | −0.016 | −0.008 | −0.007 |
| *%Diff* | - | 49 (0.28) | 55 (0.28) |
| Ammonium-N (NH$_4$-N) | −0.015 | −0.005 | −0.008 |
| *%Diff* | - | 64 (0.22) | 45 (0.26) |

**Table 3.** *Cont.*

| Manure Property | Δ Control (% Wet Basis) | Δ Pellet (% Wet Basis) | Δ Powder (% Wet Basis) |
|---|---|---|---|
| Nitrate-N ($NO_3$-N) | 0.000 | 0.000 | 0.000 |
| *%Diff* | - | 0 | 0 |
| Organic-N | −0.001 | −0.003 | 0.001 |
| *%Diff* | | −167 (0.33) | 200 (0.38) |
| Phosphorus (P) | −0.006 | −0.007 | −0.002 |
| *%Diff* | | −22 (0.41) | 61 (0.11) |
| Potassium (K) | 0.002 | −0.005 | 0.006 |
| *%Diff* | | −380 (0.15) | 280 (0.12) |
| Calcium (Ca) | −0.004 | −0.004 | −0.004 |
| *%Diff* | | 0 | 0 |
| Magnesium (Mg) | −0.003 | −0.005 | 0.000 |
| *%Diff* | | −60 (0.34) | 100 (0.09) |
| Sodium (Na) | 0.001 | −0.003 | −0.001 |
| *%Diff* | | **−550 (0.04)** | −200 (0.12) |
| Sulfur (S) | −0.003 | −0.005 | 0.000 |
| *%Diff* | | −78 (0.09) | 100 (0.22) |
| Carbon (C) | −0.003 | 0.093 | 0.013 |
| *%Diff* | | 2900 (0.15) | 500 (0.35) |
| pH | 0.003 | −0.020 | 0.007 |
| *%Diff* | | −700 (0.06) | 100 (0.47) |
| C/N ratio | −0.053 | 0.560 | 0.040 |
| *%Diff* | | 1150 (0.14) | 175 (0.36) |

## 4. Discussion

This proof-of-concept experiment showed biochar pellets have potential in short-term (up to 3 h) mitigation of $H_2S$ and $NH_3$ emissions during manure agitation but might not be as effective as biochar powder. Biochar pellet and powder treatments showed similar reductions on maximum $H_2S$ concentrations, but biochar powder showed a much higher reduction on maximum $NH_3$ concentrations. For total emissions of both $H_2S$ and $NH_3$, biochar powder showed significantly higher reductions than biochar pellets. The pH of the biochar was 5.2, which theoretically helps to retain nitrogen in liquid manure as ammonium.

Most of the biochar pellets dissolved and sunk nearly immediately after application. Some dissolved pellets fragments and powder were suspended in the manure after application (Figure A1). All biochar powder floated on top of the manure during manure agitation. Interestingly, biochar powder appeared to act as a physical barrier on the manure surface that kept most gas bubbles from being released from the liquid into the atmosphere. The apparent physical barrier behavior was confirmed (Figure A2) by applying biochar powder to the surface of deionized (DI) water followed by agitation. We attempted to further elucidate the reasons for this difference in biochar behavior by taking SEM images. Pelletization of biochar powder 'crushed' the pores with a diameter of 2–3 um (Figure 4). Thus, the pelletization made the pellets denser, less porous, and more susceptible to sinking. Prilled biochar granules generated via a process that does not necessarily involve pressure and temperature could be explored as another type of manure treatment. Biochar granules are also less dusty compared with biochar powder [33].

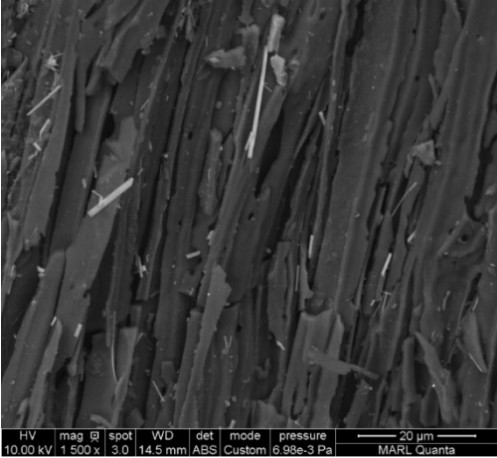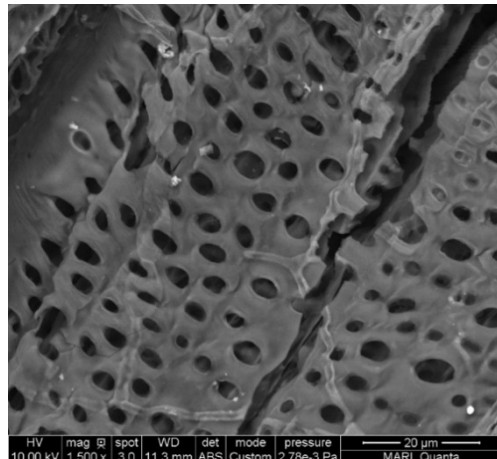

**Figure 4.** Comparison of the SEM images of biochar pellets (**left**) and biochar powder (**right**) shows morphological changes during pelletization.

The physical barrier created by the biochar can minimize the disturbed air–manure interface during manure agitation and decrease the overall mass transfer coefficient of $H_2S$. Lin et al. (2018) simulated $H_2S$ emissions during agitation and indicated that the disturbed air–manure interface induced a pH decrease, increasing the apparent overall mass transfer coefficient [34]. Thus, the biochar pH could also be exploited for specifically mitigating $H_2S$. In addition, the changes in biochar pH can potentially affect emissions of other odorants.

Results comparing the changes in manure properties confirmed that acidic biochar could mitigate $NH_3$ emissions and likely prevent nitrogen loss in the manure. Furthermore, manure treated with biochar powder showed benefits to soil health, lowered nutrient runoff risk, and the potential for agronomic benefits to corn and soybeans [22,23] shown on lab and greenhouse scales. In addition, more techno-economic analyses are warranted on the potential savings due to the nutrient retention in manure, as the average cost of anhydrous ammonia is USD \$526/ton [35].

In future research, the different binders that optimize biochar properties, including but not limited to floatability, porousness, and pH need to be evaluated. In addition, the properties of biochar vary due to the pyrolysis or torrefaction process, feedstock, and pre-treatment. Thus, biochar properties can be made specially targeted to mitigate emissions of unwanted gases. Finally, the long-term effects of biochar pellets also need to be studied.

## 5. Conclusions

This research addresses the need to develop practical mitigation technologies for the short-term release of highly toxic gases from agitated manure. Guided by the earlier success with using biochar powder to mitigate emissions, we addressed herein the concern about the hazardous nature of the fine powder. This research tested the proof-of-concept pelletized biochar application to manure surface as intrinsically safer to use to easier to apply in farm conditions. The lab-scale results showed that, while less effective than powder, the pelletized biochar can be recommended for continued research in scaled-up applications. Specific results are summarized as follows:

1. Biochar powder was significantly ($p < 0.05$) more effective than the biochar pellets.
2. Pellets reduced total $H_2S$ and $NH_3$ emissions by ~72% and ~68%, respectively ($p = 0.001$), compared with ~99% by powder ($p = 0.001$).
3. The maximum $H_2S$ and $NH_3$ concentrations were reduced from 48.1 ± 4.8 ppm and 1810 ± 850 ppm to 20.8 ± 2.95 ppm and 775 ± 182 ppm by the pellets, and to 22.1 ± 16.9 ppm and 40.3 ± 57 ppm by powder, respectively. These reductions are equivalent to reducing the maximum concentrations of $H_2S$ and $NH_3$ during the 3-h manure agitation by 57% and 57% (pellets) and 54% and 98% (powder), respectively.

4.　The changes in manure properties treated with biochar showed less loss of nitrogen and more carbon compared with the control, albeit not significant due to variability. This early observation should be further explored as the biochar treatment of manure hints at improved manure quality and, therefore, the potential for improved sustainability of the nexus of animal and crop production.

**Supplementary Materials:** The following are available online at https://www.mdpi.com/article/10.3390/atmos12070825/s1, Manure and biochar treatment properties.xlsx spreadsheet containing all manure and biochar pellet- and powder-treated manure properties.

**Author Contributions:** Conceptualization, B.C. and J.A.K.; methodology, B.C. and J.A.K.; validation, J.A.K.; formal analysis, B.C.; investigation, B.C., M.L. and P.L.; resources, J.A.K. and R.C.B.; data curation, B.C. and M.L.; writing—original draft preparation, B.C. and S.C.O.; writing—review and editing, B.C. and J.A.K.; visualization, B.C. and M.L.; supervision, J.A.K.; project administration, J.A.K.; funding acquisition, J.A.K. and R.C.B. All authors have read and agreed to the published version of the manuscript.

**Funding:** This research was partially funded by the U.S. Department of Energy—National Institute for Food and Agriculture, grant # 2018-10008-28616: 'Valorization of biochar: Applications in anaerobic digestion and livestock odor control (2018–2020, PI R.B.). Partial support came from (1) the Iowa Agriculture and Home Economics Experiment Station, Ames, Iowa. Project no. IOW05556 (Future Challenges in Animal Production Systems: Seeking Solutions through Focused Facilitation) sponsored by the Hatch Act and State of Iowa funds.

**Institutional Review Board Statement:** Not applicable.

**Informed Consent Statement:** Not applicable.

**Data Availability Statement:** The original contributions presented in the study are included in the article; further inquiries can be directed to the corresponding author.

**Conflicts of Interest:** The authors declare that the research was conducted in the absence of any commercial or financial relationships that could be construed as a potential conflict of interest.

## Appendix A

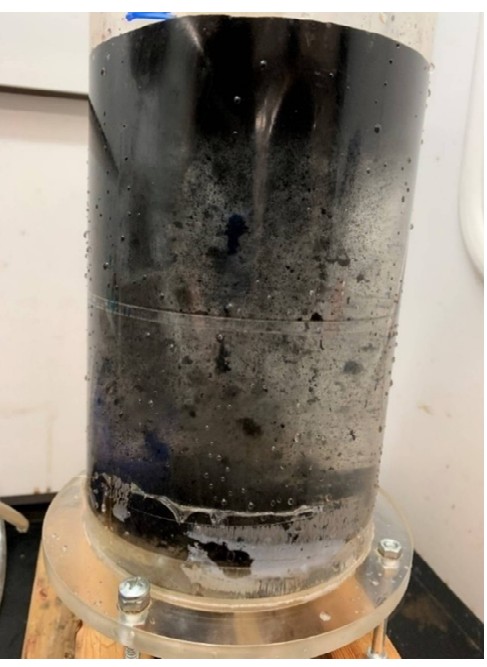

**Figure A1.** The visualization of biochar pellet dissolution during agitation demonstrated using DI water.

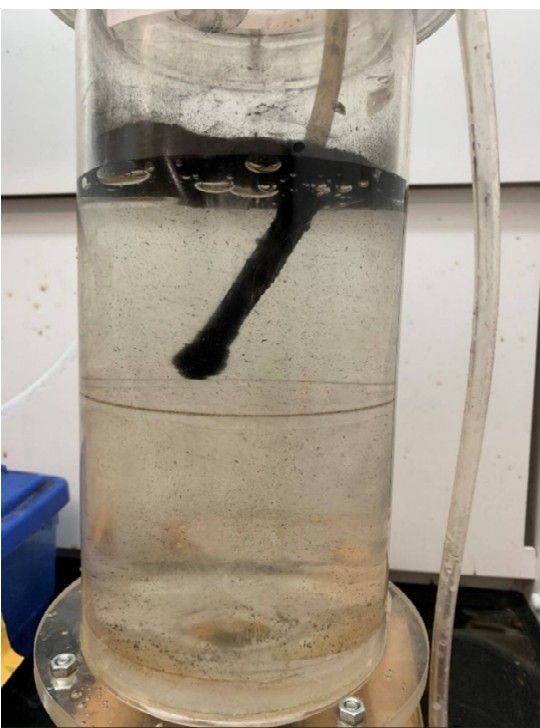

**Figure A2.** The visualization of the floating biochar powder layer that keeps the air bubbles from being released from DI water into the headspace.

**Table A1.** The average changes ($\Delta$) in manure properties from 'before' and 'after' 3-h manure agitation. A negative $\Delta$ value indicates a decrease, and a positive value indicates an increase. Moisture, mineral matter, and targeted chemicals are in units of g/L. A negative *%Diff* indicates manure properties of $\Delta$ Treatment were *%Diff* less than the manure properties of $\Delta$ Control; a positive *%Diff* indicates otherwise. The **bold** font states statistical significance.

| Manure Property (g/L) | $\Delta$ Control | $\Delta$ Pellet | $\Delta$ Powder |
|---|---|---|---|
| Moisture | 7.112 | −10.79 | 3.95 |
| *%Diff* | | −252 (0.11) | −44 (0.35) |
| Mineral Matter | −0.619 | −0.614 | 0.123 |
| *%Diff* | | 1 (0.50) | **120 (0.03)** |
| Total Nitrogen | −0.143 | −0.101 | −0.062 |
| *%Diff* | | 29 (0.39) | 57 (0.31) |
| Ammonium-N (NH$_4$-N) | −0.135 | −0.075 | −0.072 |
| *%Diff* | | 45 (0.33) | 47 (0.30) |
| Organic-N | −0.008 | −0.027 | 0.010 |
| *%Diff* | | −219 (0.31) | 214 (0.39) |
| Phosphorus (P) | −0.038 | −0.076 | −0.023 |
| *%Diff* | | −99 (0.29) | 41 (0.32) |
| Potassium (K) | 0.028 | −0.065 | 0.071 |
| *%Diff* | | −338 (0.14) | 158 (0.16) |
| Calcium (Ca) | −0.036 | −0.038 | −0.037 |
| *%Diff* | | −6 (0.48) | −2 (0.49) |
| Sodium (Na) | 0.010 | −0.036 | −0.004 |
| *%Diff* | | **−446 (0.047)** | −142 (0.12) |
| Sulfur (S) | −0.028 | −0.058 | 0.001 |
| *%Diff* | | −109 (0.106) | 104 (0.23) |
| Carbon (C) | −0.010 | 0.886 | 0.149 |
| *%Diff* | | 9346 (0.166) | 1650 (0.36) |

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
