# Peer review of "Mitigation of Acute Hydrogen Sulfide and Ammonia Emissions from Swine Manure during Three-Hour Agitation Using Pelletized Biochar"

_atmosphere, doi:10.3390/atmos12070825_

Round 1
Reviewer 1 Report
Brief summary
The aim of this paper was to compare the effectiveness of reducing H2S and NH3 from agitated swine manure using biochar pellets with biochar powder in order to protect the environment and health of both animals and workers at the farm. It did so using an well described and appropriate lab-scale approach and clearly discussed and draw conclusion from the results. The results from this paper can help improve sustainability of on farm management of swine manure, both from environmental, economic and health perspectives.
Broad comments
Abstract is clear and well structured. Minor grammatical peculiarities distract from the overall message, but it offers the reader a good introduction to the content of the manuscript.
The introduction gives a good background to the study, but is based almost exclusively on litterature in which the authors of this manuscript themselves have been involved. I would recommend the authors to include original material from the wider research community within this field of research rather than own material when describing the general state of knowledge in these questions (even if the specific question is narrow, there are related fields of research). Otherwise it becomes an introduction to the work performed within this research group, rather than the research community in this field and the research gap that this research aims to fill is not clearly illustrated.
Aim is very clear and concise. The materials and methods used for analysis are disclosed in detail and is clearly written and is easy to understand. Results are clearly described and illustrated. Discussion is also clear and to the point. It highlights the benefits and possibilities of biochar application in swine-manure. The SEM analysis added valuable insights to explain the behaviour of the biochar treatments. Conclusion section is also clear and relevant.
Specific comments
Line 79-101: These paragraphs rely exclusively on literature in which the authors have been involved. The information given is correct, but to position the study in a wider context and support the need for this study the authors should include other sources of other independent research. The lack of independent research makes it hard to judge the novelty of the research presented. (see previous comment in section above)
Line 90: Using 6 references to illustrate what is said on line 88-90 seems a little excessive, especially when they are all self-citations. There is plenty of research available on biochar production, physical properties and its applications. (see previous comment in section above)
Line 105: It is unclear what the role of methane is in this sentence: “can generate PM air pollution and potentially self-ignite [24], while methane (CH4) is generated by swine manure”. Is it a product of the applied biochar powder or a catalyst promoting self-ignition? A reference that supports this claim is needed as well.
Line 134-135: It would be more relevant and interesting to know the type of equipment used for analyzing the manure samples then the laboratory involved in the analysis.
Line 231: The %Diff given in the text don’t match that of table 3. Table 3 seems to be correct. Please correct.
Table 3 indicates changes in availability of different nutrients. It is referred to as changes in manure properties, which is somewhat unclear what it means. Many of these minerals are not volatile and will remain in the manure. They might have been adsorbed by the biochar and made unavailable for measurement detection, but they should remain in the manure+biochar mixture. When and how they will become available to plants if applied to crops in the future is an open question, but it would be good if it could be clarified with what is meant by manure properties when referring to the minerals in table 3. Is it plant available nutrients or some other fraction that is measured?
Line 258: Please explain “DI” in DI water. It is the first time the term is being used.
Line 272: “blinders” appear to be a spelling mistake. Please correct since it makes the sentence incomprehensible.
Line: 277-280: Please remove this paragraph. Left-over from guideline for authors.
Reviewer 2 Report
Comments on manuscript atmosphere-1265256
This manuscript deals with the comparison of biochar pellets and biochar powder on their effectiveness of mitigation on H2S and NH3 gases during 3-hour long swine manure agitation.
The manuscript is too simple and only are presented preliminary results. My concerns are the following:
i) Materials and methods section are too abbreviated: Missed a complete physicochemical characterization of the slurry, biochars and mixtures, lab methods used and the evolution of the parameters. Please give more detail about the gas measurements.
ii) Results section are too abbreviated: please give more detail of the results obtained.
iii) Discussion section are too abbreviated: please discuss with more detail your results and add more from literature.
iv) lines 277-280: please delete this because did not make sense.
v) Tables: Please add letters to show the differences among treatments using the comparison of means.
Round 2
Reviewer 2 Report
Comments on manuscript atmosphere-1265256_R1
My concerns about the last version of the present manuscript were explained by the authors and/or included in the revised manuscript.